# On Geometry of *p*-Adic Coherent States and Mutually Unbiased Bases

**DOI:** 10.3390/e25060902

**Published:** 2023-06-06

**Authors:** Evgeny Zelenov

**Affiliations:** Steklov Mathematical Institute, Gubkina 8, 119991 Moscow, Russia; evgeny.zelenov@gmail.com

**Keywords:** *p*-adic quantum theory, mutually unbiased bases, Hadamard matrix

## Abstract

This paper considers coherent states for the representation of Weyl commutation relations over a field of *p*-adic numbers. A geometric object, a lattice in vector space over a field of *p*-adic numbers, corresponds to the family of coherent states. It is proven that the bases of coherent states corresponding to different lattices are mutually unbiased, and that the operators defining the quantization of symplectic dynamics are Hadamard operators.

## 1. Introduction: MUBs

Mutually unbiased bases (MUBs) [1] in Hilbert space $C^{D}$ are two orthonormal bases $\{|e_{1}\rangle ,\cdots ,|e_{D}\rangle \}$ and $\{|f_{1}\rangle ,\cdots ,|f_{D}\rangle \}$ such that the square of the magnitude of the inner product between any basis states $|e_{j}\rangle$ and $|f_{k}\rangle$ equals the inverse of the dimension *D*:$$|\langle e_{j}|f_{k}\rangle {|}^{2}=\frac{1}{D},\;\forall j,k\in \left\{1,\cdots ,D\right\}.$$

Such bases have numerous applications in quantum information theory (quantum key distribution [2,3,4], quantum state tomography [5], detection of quantum entanglement [6], etcetera).

The problem is to describe the set of MUBs for an arbitrary *D*.

Within this general statement of the problem, there is a range of subtasks.

Denote by M(D) the maximum number of MUBs in $C^{D}$.

The first problem is what M(D) is equal to. In general, finding M(D) is a very difficult task; for example, M(6) has not been found to date, despite considerable efforts [7]. The answer is known when the dimension *D* is the power of a prime number, namely, $M\left(p^{n}\right)=p^{n}+1$ [8].

It is possible to obtain the following estimation [8]:$$p_{1}^{n_{1}}+1\leq M\left(D\right)\leq D+1,$$
where $D=p_{1}^{n_{1}}p_{2}^{n_{2}}\cdots p_{k}^{n_{k}},\;\;p_{1}^{n_{1}}<p_{2}^{n_{2}}<\cdots <p_{k}^{n_{k}}$ is the prime number decomposition of *D*.

The amazing thing is that this is almost all that is known at present.

The problem of finding M(D) is closely related to the well-known Winnie-the-Pooh conjecture [9]. Let us consider the Lie algebra ${\mathrm{sl}}_{D}\left(C\right)$ of D×D matrices with zero trace. The problem of decomposing this algebra into a direct sum of Cartan subalgebras that are pairwise orthogonal with respect to the Killing form is posed.

The conjecture is as follows: ${\mathrm{sl}}_{D}\left(C\right)$ is orthogonally decomposable if and only if $D=p^{n}$ for some prime *p*.

The corresponding conjecture for MUB looks like this: a complete collection of MUBs exists only in the prime power dimension *D* [10].

Let B be an orthonormal basis in $C^{D}$. Let us call a matrix *A* complex Hadamard if B and A(B) are mutually unbiased bases.

Two Hadamard matrices *A* and *C* are equivalent if there exist monomial matrices $M_{1}$ and $M_{2}$ such that the following condition is satisfied:$$A=M_{1}CM_{2}.$$
The problem is to describe the sets of equivalence classes of Hadamard matrices.

There is a complete description only for the case D≤5; for D=2,3,5 the number of Hadamard matrices is finite, and for D=4 there exists a one-dinensional family. For the case of D=6, the existence of a complex four-dimensional family of Hadamard matrices is proven [11], while for D=7, the existence of a one-dimensioin family is proved [12].

There are difficulties with the definition of mutually unbiased bases in the case of an infinite-dimensional Hilbert space [13]. In this paper we provide such a definition. Despite its seeming naivety, the definition mentioned above naturally arises in the context of *p*-adic quantum mechanics.

It should be noted here that the above brief overview and bibliographic references have no claim to being complete, as many important and interesting articles are not mentioned.

## 2. *p*-Adics Numbers

A few words about *p*-adic numbers are required in order to introduce the necessary notation. For more information about *p*-adic numbers, see for example [14].

We first fix a prime number *p*. Any rational number x∈Q is uniquely representable as
$$x=p^{k}\frac{m}{n},\;k,m,n\in Z,\;n>0,\;p∤m,\;p∤n.$$
Let us define the norm ${\left|\cdot \right|}_{p}$ on Q by the formula ${\left|x\right|}_{p}=p^{-k}$; completion of the field of rational numbers with this norm is the field $Q_{p}$ of *p*-adic numbers. The *p*-adic norm of a rational integer n∈Z is always less than or equal to one, ${\left|n\right|}_{p}\leq 1$, and the completion of rational integers Z with the *p*-adic norm is denoted by $Z_{p}$. $Z_{p}=x\in Q_{p}:{\left|x\right|}_{p}\leq 1$, that is, it is a disk of a unit radius.

For the *p*-adic norm, the strong triangle inequality holds:$${\left|x+y\right|}_{p}\leq {\max\{|x|}_{p}{,|y|}_{p}\}.$$

The non-Archimedean norm defines the totally disconnected topology on $Q_{p}$ (i.e., the disks are open and closed simultaneously).

Two disks either do not intersect, or one lies in the other.

Locally constant functions are continuous, for example,
$$h_{Z_{p}}\left(x\right)=\left\{\begin{array}{c} 1,x\in Z_{p},\\ 0,x\notin Z_{p}, \end{array}\right.$$
is a continuous function.

$Q_{p}$ is Borel isomorphic to the real line R. The shift-invariant measure dx on $Q_{p}$ (the Haar measure) is normalized in such a way that $\int_{Z_{p}}dx=1$.

For any nonzero *p*-adic number, the canonical representation holds:$$Q_{p}∋x=\sum_{k=-n}^{+\infty }x_{k}p^{k},\;n\in Z_{+},\;x_{k}\in \left\{0,1,\cdots ,p-1\right\}.$$
Using the canonical representation, we define the integer ${\left[x\right]}_{p}$ and fractional ${\left\{x\right\}}_{p}$ parts of the number $x\in Q_{p}$:$$\underset{{\left\{x\right\}}_{p}}{\underset{︸}{p^{-n}x_{-n}+p^{-n+1}x_{-n+1}+\cdots +p^{-1}x_{-1}}}+\underset{{\left[x\right]}_{p}}{\underset{︸}{x_{0}+px_{1}+\cdots +p^{k}x_{k}+\cdots }}.$$
The following function, which takes values in a unit circle T in C, is the additive character of the field of *p*-adic numbers:$$\chi_{p}\left(x\right)=\exp\left(2\pi i{\left\{x\right\}}_{p}\right),\;\;\chi_{p}\left(x+y\right)=\chi_{p}\left(x\right)\chi_{p}\left(y\right).$$
The *p*-Adic integers $Z_{p}$ form a group with respect to addition (a consequence of the non-Archimedean norm), and their group is a profinite (procyclic) group. This is the inverse limit of finite cyclic groups $Z/p^{n}Z,\;n\in N$:$$Z/pZ⟵\cdots ⟵Z/p^{n}Z⟵Z/p^{n+1}Z⟵\cdots .$$
Consider the group ${\hat{Z}}_{p}$ of characters $Z_{p}$. This group has the following form:$${\hat{Z}}_{p}=Q_{p}/Z_{p}=Z\left(p^{\infty }\right)=\left\{\exp(2\pi im/p^{n}),\;m,n\in N\right\}.$$
This is the Prüfer group. It is a direct limit of finite cyclic groups (i.e., quasicyclic) of order $p^{n}$:$$Z/pZ\to Z/p^{2}Z\to \cdots \to Z/p^{n}Z\to \cdots .$$

## 3. Representations of CCR: Coherent States

Let $V=Q_{p}^{2}$ be a two-dimensional vector space over $Q_{p}$ and let Δ be a non-degenerate symplectic form on this space.

Let H be a separable complex Hilbert space. A map *W* from *V* to a set of unitary operators on H satisfying the condition
$$W\left(u\right)W\left(v\right)=\chi_{p}\left(\Delta \left(u,v\right)\right)W\left(v\right)W\left(u\right),\;u,v\in V$$
is called a representation of canonical commutation relations (CCR). Furthermore, we require both continuity in a strong operator topology and irreducibility. When these conditions are met, such a representation is unique up to unitary equivalence.

The *p*-Adic integers $Z_{p}$ form a ring. Let *L* be a two-dimensional (compact) $Z_{p}$-submodule of the space *V*. Such submodules will be called lattices.

On the set of lattices, we introduce the operations ∨ and ∧:$$L_{1}\vee L_{2}=L_{1}+L_{2}=\left\{z_{1}+z_{2},\;z_{1}\in L_{1},\;z_{2}\in L_{2}\right\},$$
$$L_{1}\wedge L_{2}=L_{1}\cap L_{2}.$$
We additionally define the involution ∗:$$L^{*}=\left\{z\in V:\Delta \left(z,u\right)\in Z_{p}\forall u\in L\right\}.$$
It is easy to see that ${\left(L_{1}\wedge L_{2}\right)}^{*}=L_{1}\vee L_{2}$. The lattice *L* that is invariant with respect to the involution is called self-dual, $L=L^{*}$.

We normalize the measure on *V* in such a way that the volume of a self-dual lattice is equal to one. The symplectic group $Sp\left(V\right)=SL_{2}\left(Q_{p}\right)$ acts transitively on the set of self-dual lattices.

By L, we denote the set of self-dual lattices. On the set L, we define the metric *d* by the formula
$$d\left(L_{1},L_{2}\right)=\frac{1}{2}\log\#\left(L_{1}\vee L_{2}/L_{1}\wedge L_{2}\right),$$
where log further denotes the logarithm to the base *p* and # is the number of elements of the set.

**Example** **1.**
*Let {e,f} be a symplectic basis in V,Δ(e,f)=1. Then, the lattices*

$$L_{1}=Z_{p}e⊕Z_{p}f,\;\;L_{2}=p^{n}Z_{p}e⊕p^{-n}Z_{p}f$$

*are self-dual. If n≥0, then*

$$\begin{array}{cc} L_{1}\wedge L_{2}=p^{n}Z_{p}e⊕Z_{p}f,\;\;\;L_{1}\vee L_{2} & =Z_{p}e⊕p^{-n}Z_{p}f,\\ & d\left(L_{1},L_{2}\right)=\frac{1}{2}\log\#\left(L_{1}\vee L_{2}/L_{1}\wedge L_{2}\right)=\frac{1}{2}\log p^{2n}=n. \end{array}$$



Note that such a basis exists for any pair of self-dual lattices.

The set of self-dual lattices can be represented as a graph. The distance *d* takes values in the set of non-negative integers. The vertices of the graph are elements of the set L, and the edges are pairs of self-dual lattices $\left\{L_{2},L_{2}\right\}:d\left(L_{1},L_{2}\right)=1$.

The graph of self-dual lattices is constructed according to the following rule. Let $K_{p+1}$ denote a complete graph with p+1 vertices. The countable family of copies of the graph $K_{p+1}$ is glued together in such a way that each vertex of each graph in this family belongs to exactly p+1 graphs $K_{p+1}$.

By replacing each complete graph $K_{p+1}$ with a star graph $S_{p+1}$, we obtain a Bruhat–Tits tree.

We now proceed with the construction of the vacuum vector. Let us choose a self-dual lattice L∈L and consider the operator
$$P_{L}=\int_{L}dzW\left(z\right).$$

**Lemma** **1.**
*The $P_{L}$ operator is a one-dimensional projection.*


Indeed, we have
$$P_{L}^{2}=\int_{L}dzW\left(z\right)\int_{L}dz^{\prime }W\left(z^{\prime }\right)=\int_{l}dz\int_{L}dz^{\prime }W\left(z+z^{\prime }\right)=\int_{L}dzW\left(z\right)=P_{L}.$$
The one-dimensionality of the projection $P_{L}$ directly follows from the irreducibility of the representation *W*.

Our desired vacuum state will be this projection. We fix the notation $P_{L}=|0_{L}\rangle \langle 0_{L}|$.

**Definition** **1.**
*The family of states $\{|z_{L}\rangle =W\left(z\right)|0_{L}\rangle ,\;z\in V\}$ in H is said to be the system of (L-)coherent states.*


We denote by $h_{L}$ the indicator function of the lattice *L*,
$$h_{L}\left(z\right)=\left\{\begin{array}{c} 1,z\in L,\\ 0,z\notin L. \end{array}\right.$$

**Theorem** **1.**
*Coherent states satisfy the following relation:*

$$|\langle z_{L}|z_{L}^{\prime }\rangle |=h_{L}\left(z-z^{\prime }\right).$$

*In other words, the coherent states $|z_{L}\rangle \langle z_{L}|$ and $|z_{L}^{\prime }\rangle \langle z_{L}^{\prime }|$ coincide if $z-z^{\prime }\in L$ and are orthogonal otherwise.*


Let $u=z-z^{\prime }$; then,
$$|\langle z_{L}|z_{L}^{\prime }\rangle \left|=\right|\chi_{p}\left(1/2\Delta \left(z,u\right)\right)\langle 0_{L}|W\left(u\right)0_{L}\rangle \left|=\right|\langle 0_{L}|W\left(u\right)0_{L}\rangle |.$$
If u∈L, then the statement in the theorem follows from the definition of a vacuum vector. If u∉L, then by virtue of the self-duality of the lattice *L*, there exists v∈L such that $\chi_{p}\left(\Delta \left(u,v\right)\right)\neq 1$. We then have
$$\langle 0_{L}|W\left(u\right)0_{L}\rangle =\langle 0_{L}|W\left(-v\right)W\left(u\right)W\left(v\right)0_{L}\rangle =\chi_{p}\left(\Delta \left(u,v\right)\right)\langle 0_{L}|W\left(u\right)0_{L}\rangle ,$$
which is true only if $\langle 0_{L}|W\left(u\right)0_{L}\rangle =0$.

Therefore, non-matching (and pairwise orthogonal) coherent states are parametrized by elements of the set $V/L={\left(Q_{p}/Z_{p}\right)}^{2}≅Z\left(p^{\infty }\right)\times Z\left(p^{\infty }\right).$ This makes the following modification of Definition 1 natural.

**Definition** **2.**
*The set $\{|\alpha_{L}\rangle =W\left(\alpha \right)|0_{L}\rangle ,\;\alpha \in V/L\}$ is said to be the basis of p-adic (L-)coherent states.*


**Remark** **1.**
*The CCR representations are closely related to the representations of the Heisenberg group. In the language of representation theory, p-adic coherent states are nothing other than coherent states for the *p*-adic Heisenberg group.*


## 4. Main Result

Let $L_{1}$ and $L_{2}$ be a pair of self-dual lattices $d(L_{1},L_{2})\geq 1$.

It turns out that the corresponding bases of $L_{1}$-coherent and $L_{2}$-coherent states are mutually unbiased on finite-dimensional subspaces of dimension $p^{d(L_{1},L_{2})}$.

**Theorem** **2.**
*For bases of $L_{1}$-coherent and $L_{2}$-coherent states $\{|\alpha_{L_{1}}\rangle ,\;\alpha \in V/L_{1}\}$ and $\{|\beta_{L_{2}}\rangle ,\;\beta \in V/L_{2}\}$, the following formula is valid:*

$$|\langle \alpha_{L_{1}}|\beta_{L_{2}}\rangle {|}^{2}=p^{-d(L_{1},L_{2})}h_{L_{1}\vee L_{2}}\left(\alpha -\beta \right).$$



The above theorem means the following. Our Hilbert space for representation of CCR H decomposes into an orthogonal direct sum of finite-dimensional subspaces of the same dimension $p^{d(L_{1},L_{2})}$:$$H=\underset{a\in V/(L_{1}\vee L_{2})}{⨁}H_{a},\;\dim H_{a}=p^{d(L_{1},L_{2})}.$$
In each of these subspaces, the sub-bases of $L_{1}$-coherent and $L_{2}$-coherent states are mutually unbiased.

Let us now prove Theorem 2.

The following formula is valid:(1)$$|\langle 0_{L_{1}}|W\left(\beta \right)0_{L_{2}}\rangle |=\left\{\begin{array}{c} |\langle 0_{L_{1}}|0_{L_{2}}\rangle |,\beta \in L_{1}\vee L_{2},\\ 0,\beta \notin L_{1}\vee L_{2}. \end{array}\right.$$

Let $\beta \in L_{1}\vee L_{2}$; then, $\beta =\beta_{1}+\beta_{2},\;\beta_{1}\in L_{1},\;\beta_{2}\in L_{2}$ and
$$|\langle 0_{L_{1}}|W\left(\beta \right)0_{L_{2}}\rangle \left|=\right|\langle W\left(-\beta_{1}\right)0_{L_{1}}|W\left(\beta_{2}\right)0_{L_{2}}\rangle \left|=\right|\langle 0_{L_{1}}|0_{L_{2}}\rangle |.$$
If $\beta \notin L_{1}\vee L_{2}$, then there exists $\gamma \in L_{1}\wedge L_{2}={\left(L_{1}\vee L_{2}\right)}^{*}$ such that $\chi_{p}\left(\Delta (\gamma ,\beta \right)\neq 1$ and
$$\langle 0_{L_{1}}|W\left(\beta \right)0_{L_{2}}\rangle =\langle W\left(\gamma \right)0_{L_{1}}|W\left(\beta \right)W\left(\gamma \right)0_{L_{2}}\rangle =\chi_{p}\left(\Delta (\gamma ,\beta \right)\langle 0_{L_{1}}|W\left(\beta \right)0_{L_{2}}\rangle .$$
From the latter equality, it obviously follows that $\langle 0_{L_{1}}|W\left(\beta \right)0_{L_{2}}\rangle =0,\;\beta \notin L_{1}\vee L_{2}$.

Now let us use formula (Equation 1) and the Parseval–Steklov identity:(2)$$1=\sum_{\beta \in V/L_{2}}{\left|\langle 0_{L_{1}}|W\left(\beta \right)0_{L_{2}}\rangle \right|}^{2}={\left|\langle 0_{L_{1}}|0_{L_{2}}\rangle \right|}^{2}\sum_{\beta \in L_{1}\vee L_{2}/L_{2}}1={\left|\langle 0_{L_{1}}|0_{L_{2}}\rangle \right|}^{2}p^{d(L_{1},L_{2})}.$$

The following equation follows from Formula (Equation 2):(3)$${\left|\langle 0_{L_{1}}|0_{L_{2}}\rangle \right|}^{2}=p^{-d(L_{1},L_{2})}.$$
Taking into account Formula (Equation 1) and the equality (Equation 3), we obtain a proof of Theorem 2.

In the case of $d(L_{1},L_{2})=1$, the subspaces $H_{a},\;a\in V/\left(L_{1}\vee L_{2}\right)$ have dimension *p*. As can be seen from the construction of the graph of lattices, there are exactly p+1 pieces of self-dual lattices with unit pairwise distances (the complete graph $K_{p+1}$). These lattices define a complete set of MUB in each subspace $H_{a}$.

In the case of $d(L_{1},L_{2})=2$, the subspaces $H_{a},\;a\in V/\left(L_{1}\vee L_{2}\right)$ have dimension $p^{2}$. As can be seen from the construction of the graph of lattices, there are exactly p(p+1) pieces of self-dual lattices lying at a distance of 2 from lattice $L_{1}$. The bases corresponding to these lattices are not mutually unbiased. However, among this set there are families consisting of p+1 pieces of mutually unbiased bases. These bases saturate the entropic uncertainty relations [15,16].

**Remark** **2.**
*Instead of the field $Q_{p}$, we can consider its algebraic extension of degree n. Such extensions exist for any n. In this case, the elementary building block of the lattice graph will be the complete graph $K_{p^{n}+1}$, which has vertex $p^{n}+1$. The coherent state bases corresponding to the vertices of this graph form a complete set of mutually unbiased bases in $p^{n}$-dimensional space.*


The theorem makes the following definitions natural.

**Definition** **3.**
*Let H be an infinite-dimensional Hilbert space. The orthonormal bases $\left\{|e_{i}\rangle \right\}$ and $\left\{|f_{j}\rangle \right\}$ are mutually unbiased if there exists a decomposition*

$$H=⊕H_{k},\;\dim H_{k}=n_{k}<\infty ,$$

*such that the sub-bases $\{|e_{i}\rangle {\}|}_{H_{k}}$ and $\{|f_{j}\rangle {\}|}_{H_{k}}$ are mutually unbiased for all k.*


We now make two important remarks. First, the above definition assumes that the bases are divided into finite blocks of size $n_{k}$, each of which forms a sub-base in the corresponding subspace $H_{k}$. Second, in the case under consideration, that is, the representation of CCR over a field of *p*-adic numbers with the dimension of the subspaces being $H_{k}$, there are powers of *p*. This construction can be extended to the case of CCR over Vilenkin groups, in which case the above dimensions can be arbitrary natural numbers.

## 5. p-Adic Dynamics: Hadamard Operators

The proposed definition of mutually unbiased bases for the case of an infinite-dimensional Hilbert space makes it possible to introduce the concept of the Hadamard operator for such spaces in a similar way.

**Definition** **4.**
*The operator A in the Hilbert space H is called the Hadamard operator if for some orthonormal basis $\left\{|e_{i}\rangle \right\}$ in H the bases $\left\{|e_{i}\rangle \right\}$ and $A(\{|e_{i}\rangle \})$ are mutually unbiased.*


In other words, the Hadamard operator is provided by an infinite block-diagonal matrix with diagonal blocks that are ordinary finite Hadamard matrices.

It turns out that the dynamics of *p*-adic quantum systems are determined by Hadamard operators. More detailed information about *p*-adic quantum theory can be found in [17,18]. The dynamics of a classical system are provided by a linear symplectic transformation g∈Sp(V) of the phase space *V*. A one-parameter family $g_{t}$ of such transformations can be specified, in which case the parameter $t\in Q_{p}$ is interpreted as time. For example, the dynamics of a free particle of unit mass are provided by the family $g_{t},\;t\in Q_{p}$, which in some fixed basis of space *V* has the foollowing form:$$g_{t}=\left(\begin{array}{cc} 1 & t\\ 0 & 1 \end{array}\right),\;t\in Q_{p}.$$
If the dynamics of a classical system are determined by the action of a symplectic group on the phase space, then the dynamics of the corresponding quantum system are provided by the so-called metaplectic representation of the symplectic group in the Hilbert space of the representation of CCR. The existence of such a representation follows from the uniqueness of irreducible representations of CCR.

Let (W,H) be a representation (irreducible) of CCR and g∈Sp(V). Then, by virtue of the uniqueness of the representation, the representations (W,H) and $\left(W_{g},H\right),\;W_{g}\left(z\right)=W\left(gz\right),\;z\in V$ are unitarily equivalent, that is, there is a unitary operator U(g) satisfying the condition
$$U\left(g\right)W\left(z\right)=W_{g}\left(z\right)U\left(g\right),\;z\in V.$$
The operators U(g),g∈Sp(V) define a metaplectic representation of Sp(V).

**Theorem** **3.**
*Let L be a lattice in V such that d(L,gL)≥1. Then, U(g) is the Hadamard operator for bases ${\{|\alpha \rangle }_{L}\}$ and ${\{|\beta \rangle }_{gL}\}$.*


As mentioned above, the symplectic group acts transitively on the set of self-dual lattices. Thus, if a self-dual lattice *L* is given, its image gL under the action of the symplectic transformation *g* is a self-dual lattice as well. Thus, the validity of Theorem 3 follows from Theorem 2 for a pair of lattices *L* and gL. 

## Data Availability

No new data were created or analyzed in this study. Data sharing is not applicable to this article.

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
