# Peer review of "On Geometry of p-Adic Coherent States and Mutually Unbiased Bases"

_entropy, 2023, doi:10.3390/e25060902_

Round 1

Reviewer 1 Report

This paper considers coherent states for the representation of Weyl commutation relations
over a field of p-adic numbers. A geometric object, a lattice in vector space over a field of p-adic
numbers, corresponds to the family of coherent states. It is proved that the bases of coherent states
corresponding to different lattices are mutually unbiased, and the operators defining the quantization
of symplectic dynamics are Hadamard operators.

This paper is well written, but I suggest that the paper should discuss possible applications for the special coherent states,  for example, if their results can be used to calculate the analytical entanglement in the model of Ref. [Coherent State Control to Recover Quantum
Entanglement and Coherence, Entropy 2019, 21, 917; doi:10.3390/e21100917]? If can, how to calculate?

Author Response

Thank you so much for working on the review. 

You have raised a very interesting question. The question is complicated, and I don't have an answer to it yet. Nevertheless, thanks for the great idea of trying to construct an analogue of the Jaynes-Cummins model using p-adic coherent states. 

There are two main difficulties here. The first is to describe coherent states for a multimode (or at least two-mode) system. That's what I'm working on now. In contrast to the single—mode system discussed in the article, in the multimode case, more complex geometric objects arise - the Bruhat-Tits buildings. The second difficulty lies in the fact that the representation of the Heisenberg group over p-adic numbers have no generators (due to topology of the field of p-adic numbers), therefore there are no creation and annihilation operators. Thus, we are forced to work with evolution operators instead of  Hamiltonians, this creates significant methodological and technical difficulties.

Reviewer 2 Report

I recommend to publish the paper after some minor typos and bugs are fixed during proofreading.   The list  of these follows:

  1. line 6: delete the first of «mutually»
  2. line 43: I’d rather use «claim» than «pretend» here
  3. two lines below line  57:   something is missing before \ni in the formula
  4. line 58, 4 lines up:  something is missing in the formula, before and after / 
  5. line 58, one line up:  here must be \mathbb Z rather than \mathbb Z_p everywhere in the formula
  6. line 59:   missed \mathbb Q_p  after «over»
  7. line 60, one line below:  missed  W  before «from»
  8. line 64: missed \mathbb  Z_p twice,  after «a ring» and before «—submodule»
  9. line 69: missed symbol in parenthesis, SL_2(?)
  10. line 87, two lines below:  missing |  before right = in the formula
  11. line 90:  missing  symbols in the formula, (?/?)^2
  12. line 136, one line up:  something is missed in the formula,  after \in
  13. line 141,   3 lines up: missed operator symbol after «operator»
  1. line 43: I’d rather use «claim» than «pretend» here

English in the rest of the paper  is OK^ at my view.

Author Response

Many thanks for the thorough work on the review. I took into account all the comments and corrected the typos.

Reviewer 3 Report

p-adic quantum theory, initiated in some problems of field theory, provides a new approach to the description of fundamental physical regularities and is currently an actively developing area of mathematical physics.

The peer-reviewed paper studies coherent states to represent the Weyl commutation relations over a field of p-adic numbers. Theorems on the properties of coherent states are proven, which are essential.

The article is written clearly, the purpose of the work is sufficiently motivated, the results obtained contribute to p-adic quantum theory.

In my opinion, the paper can be accepted for publication in Entropy in its present form.

Author Response

Thank you so much for working on the review.

Round 2

Reviewer 1 Report

I agree their answer